# Lysozyme: A Natural Product with Multiple and Useful Antiviral Properties

**DOI:** 10.3390/molecules29030652

**Published:** 2024-01-30

**Authors:** Alberta Bergamo, Gianni Sava

**Affiliations:** Department of Life Sciences, University of Trieste, 34127 Trieste, Italy; abergamo@units.it

**Keywords:** lysozyme, antiviral activity, mechanism of action, disinfectant

## Abstract

Lysozyme, especially the one obtained from hen’s egg white, continues to show new pharmacological properties. The fact that only a few of these properties can be translated into therapeutic applications is due to the lack of suitable clinical studies. However, this lack cannot hide the evidence that is emerging from scientific research. This review for the first time examines, from a pharmacological point of view, all the relevant studies on the antiviral properties of lysozyme, analyzing its possible mechanism of action and its ability to block viral infections and, in some cases, inhibit viral replication. Lysozyme can interact with nucleic acids and alter their function, but this effect is uncoupled from the catalytic activity that determines its antibacterial activity; it is present in intact lysozyme but is equally potent in a heat-degraded lysozyme or in a nonapeptide isolated by proteolytic digestion. An analysis of the literature shows that lysozyme can be used both as a disinfectant for raw and processed foods and as a drug to combat viral infections in animals and humans. To summarize, it can be said that lysozyme has important antiviral properties, as already suspected in the initial studies conducted over 50 years ago, and it should be explored in suitable clinical studies on humans.

## 1. Introduction

Lysozyme is a term used to describe dozens of peptides found in the animal and plant world that share a highly conserved chemical structure, albeit with some, sometimes significant, differences between them [1]. Human lysozyme is, of course, one of the most studied lysozymes [2,3], followed by hen egg white lysozyme (HEWL), the only lysozyme also available in a pharmaceutical form and used for therapeutic purposes (Figure 1) [4,5,6]. Since Sir Alexander Fleming described its biological properties and medical potential [7], it has been present in research laboratories for over 100 years. In 1953, in a scientific meeting at the New York Academy of Medicine, Meyer highlighted the biological properties of this enzyme, pointing out its antibacterial activity and, in particular, its biological and metabolic functions [8]. Although it has not achieved the success we would have expected, e.g., as an antibacterial drug [9,10,11], apart from its use in the food industry [12], it has become one of the most studied molecules in the biological and medical sciences and one of the most widely used models in biochemistry [13,14,15].

A limitation of its potential use stems from its chemical nature (a small peptide of 14-18 kD, depending on its source) as a potentially allergenic substance (a phenomenon that is further enhanced in the case of a lysozyme extracted from hen egg white, which may contain traces of egg contaminants), a property that has been extensively studied [16], and references are cited herein. Nevertheless, studies on the chemistry, biochemistry, physiology, and pharmacology of lysozyme have led to research that has emphasized its potential use in therapy for various conditions for which both direct and indirect effects are expected. Thus, the antibacterial activity of lysozyme could contribute to its immunomodulatory effects, although there are data that suggest the modulation of the immune system also occurs when the molecule loses its ability to generate the peptidoglycan fragments thought to be responsible for stimulating immunity (for a full account of this property, see [17]). According to the expectations of its discoverer, lysozyme was supposed to be a natural antibacterial agent, and many studies have emphasized its role as the first barrier to control the entry of pathogenic microorganisms into the organism [7,9,10,11]. An interesting paper on the antibacterial activity of lysozyme comes from Ragland and Criss who emphasized the crucial role of this enzyme in the so-called natural immunity and its fascinating function as an immunomodulator for coping with infections [18].

**Figure 1 molecules-29-00652-f001:**
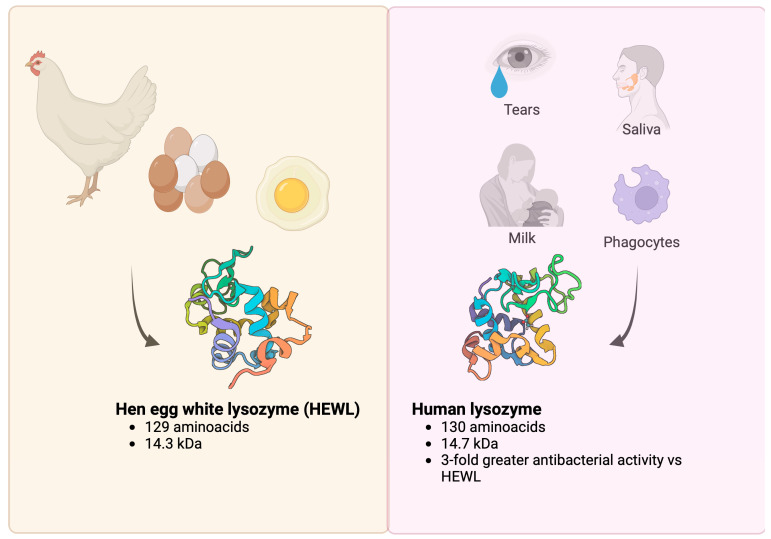
The most studied lysozymes. Human lysozyme and HEWL are the most known and characterized among c-type lysozymes. (Crystal structure of hen egg white lysozyme: 5B1F [19]; crystal structure of human lysozyme: 7XF6 [20].)

In the present review, we would like to highlight another therapeutic target identified by some researchers who have investigated the potential applications of this molecule, namely, antiviral activity. A short report published in Nature in 1959 presents a series of studies with some contradictory results on the antiviral role of lysozyme and concludes that the different types of viruses (herpes simplex, herpes zoster, warts, condylomata acuminata, aphthosis, and vaccinia viruses) are sensitive to this agent, but that further confirmatory studies need to be carried out [21]. The indirect evidence for the role of lysozyme in combating viruses comes from the observation that ocular herpes simplex infections were associated with a decrease in lysozyme levels in the infected eye compared to the healthy eye of the same patient [22] and in general to the eyes of healthy people [23]. Further evidence for the role of lysozyme content in reducing viral infections comes from a study on *Bombyx morii*, an insect in which viral infections are controlled by a significant increase in the overexpression of C-lysozyme [24].

In 1959, lysozyme was administered parenterally, with a high risk of causing allergic reactions in patients; this route of administration was finally discontinued in the following years. It should be recalled that lysozyme has good intestinal absorption, as evidenced by a study highlighting absorption mediated by endocytic and paracellular pathways in the proximal intestine [25], and that the preparations of lysozyme for oral use are available on the market (Lysozyme SPA) (Figure 2).

The absorption of lysozyme by the upper intestine was further investigated and shown to be dependent on an endocytic pathway [26]. This study also demonstrated that the presence of megalin, the receptor for its endocytosis in the lower intestine, is much less involved in the overall absorption of the peptide. The occurrence of possible allergic reactions in particularly sensitive individuals cannot be ruled out even when lysozyme is taken orally. In any case, the European Food Safety Authority recently stated that the amounts of lysozyme (lysozyme from hen egg white, peptidoglycan-N-acetylmuramoyl hydrolase; EC 3.2.1.17) that can be introduced as a preservative into foods are significantly lower than the amount of lysozyme that a person ingests with the normal consumption of eggs and should therefore be considered safe [27]. Lysozyme can therefore be considered, at least in terms of pharmacokinetics, as a complete medicinal product that can be taken orally and is therefore useful for the treatment of systemic diseases. This aspect is fundamental to substantiate the potential therapeutic applications described for lysozyme, whether hypothesized or proven (please refer to the review [28]).

The interest and scientific impact of the Ferrari et al. study [21] fell short of expectations in terms of rapid and widespread antiviral application in humans, and what we discuss below about the antiviral activity of lysozyme is based on research results from the last 20 years, i.e., more than 50 years after its first publication. However, the main objective of the present review is not to analyze a structure–activity relationship (QSAR) but rather to present the data and information supporting the antiviral activity of lysozyme from biological and pharmacological points of view. Therefore, the review presents, analyzes, and discusses each published work taken into consideration without any priority given to one over another. In order to do that, we have carefully examined PubMed and selected the papers that were extracted using different combinations of the keywords, such as lysozyme, antiviral, activity, therapy, treatment, and mechanism of action, with particular attention, though not in an exclusive way, given to the papers published in the last 20 years.

## 2. Lysozyme Modulates Nucleic Acid Activity

Lysozyme is able to interact with nucleic acids, as shown by the results of gel electrophoresis, enzymatic activity, and co-precipitation studies [29], which demonstrate that both HEWL and human lysozyme are able to bind with DNA and RNA. These studies emphasize that DNA requires specific conformations, and that the interaction is physical and electrostatic in nature, as lysozyme is easily separated from a nucleic acid (Figure 3). From a study designed to demonstrate that ATP binds to proteins, modifying their structure and modulating their functions, it appears that lysozyme has multiple binding sites, including one known high-affinity nucleic acid binding site and five non-specific, previously unknown sites [30]. Although this study aimed to demonstrate the modifications that ATP and not TTP exerts on the conformation of proteins, if we read the results as a function of lysozyme, we can also see how it has the structural and kinetic features to interact with nucleic acids. However, these data do not allow us to determine if there is an effect on the structure of DNA (or RNA), such as altered curvature, but allow us to hypothesize that lysozyme may have a role as a regulator of DNA and RNA in the cytoplasm and the extracellular environment [29]. This hypothesis, which is supported by the results of studies by Lee-Huang et al. [31], demonstrating the protective effect of lysozyme on HIV infection, states that lysozyme is actively involved in the processes of viral transcription and replication [29] as an original part of the body’s defenses [32], which indicates that the interaction of lysozyme with DNA molecules can disrupt DNA replication, modulate gene expression, and influence bacterial infections, in addition to HIV viruses.

On the other hand, lysozyme, with its strong positive charge (+8e), its compact structure, and depending on its concentration, is able to alter the charge and electrophoretic mobility of DNA [35,36], a fundamental event that plays an important role in cellular processes such as transcription, translation, and cell division. However, these biological properties may also be the result of specific interactions, in addition to electrostatic interactions, which also play an important role in the binding of lysozyme with nucleic acids, as confirmed with surface plasmon resonance spectroscopy studies that identified DNA-binding motifs at the N- and C- terminals of lysozyme [33].

Studies with T7 lysozyme, a lysozyme that exerts its catalytic activity at an amide bond rather than a glycosidic bond [37], have shown that it is able to control the activity of RNA polymerase by inhibiting both the initiation and continuation of its activity [34]. Although this study was performed with a purified material of bacteriophage origin, the signal is evident in its ability to modulate RNA synthesis, and this effect can have devastating consequences for viruses, especially those that rely on RNA. The inhibition of RNA polymerase leads to the inhibition of transcription [38,39], which ultimately leads to the control of infectivity [40].

The possibility that lysozyme can bind nucleic acids arises from studies that have demonstrated the similarity of this peptide to histones [29]. The recent study by Liu et al. [41] builds on these observations and uses PCR to investigate the effects of lysozyme on the replication and transcription of nucleic acids and finds that the inhibitory effect caused by lysozyme is strong and mediated by the interaction of lysozyme with RNA polymerase, and it concludes that this activity could explain the observed antiviral effect. The fact that lysozyme is able to bind peptides is nothing new. For example, two lysozymes from hen and turkey egg whites were analyzed using X-ray diffraction and were found capable of binding with the antibody fragment 1F9 [42]. If the crystalline structure is compared with a model of the single-chain Fv fragment 1F9 complexed with HEWL, it can be seen that there is a collision between Asp101 in the lysozyme and Trp98 of the complementarity-determining region H3 of the variable domain of the heavy chain.

## 3. Chemical Structure and Anti-Viral Activity

Studies on the interactions of lysozyme with nucleic acids suggest that lysozyme is able to influence DNA replication by interacting directly with it, but it is also able to modulate gene expression by interacting with RNA polymerase. The obvious question to ask now is what role does the chemical structure of lysozyme play in its antiviral activity? In the previous section, we saw that both HEWL and human lysozyme exert largely the same effects on nucleic acids. However, it is known that human lysozyme, which has a Tyr62, has at least 2–4 times higher lytic (antibacterial) activity than HEWL, which has a Trp at the same site [43]. It is therefore likely that the lytic activity, i.e., the antibacterial activity, and the antiviral activity, i.e., the interaction/modulation with/of nucleic acids, are based on different principles. The antiviral activity of lysozyme is attributed to several lysozymes, including HEWL and human lysozyme (see the review [44]), and is attributed to a mechanism of action independent of muramidase activity, which is related to the positive charge of the molecule [45]. The indirect evidence for the antiviral activity of lysozyme comes from a study showing the presence of this peptide in the preparations of human chorionic gonadotropin [31]. One of the isolated peptides has a sequence and muramidase activity comparable to those of human lysozyme and HEWL and showed an antiviral activity without a cytotoxic effect on eukaryotic cells in tests with mononuclear cells exposed to the HIV virus type 1, as shown by the study of ^3^H-thymidine incorporation. However, the mechanism of the antiviral activity was not clarified in this study. 

An attempt to modify the chemical structure of lysozyme to achieve antiviral activity was made by Oeverman et al. [46], who introduced a hydrophobic and negatively charged function (3-hydroxyphthalic anhydride) into the lysozyme molecule from hen egg white. The modified lysozyme molecule showed a remarkable in vitro antiviral activity in a herpes simplex virus model, both in terms of preventing cell infection and, most importantly, and inhibiting virus replication when the drug was added to cells previously infected with the virus (the IC_50_ decreased from 170 mg/mL to 6 mg/mL). The same study also showed that the proteolytic digestion of lysozyme with trypsin, chymotrypsin, and pepsin produces fragments with no or significantly lower antiviral activity, often only when used as a mixture, and almost always accompanied by some cytotoxicity to the eukaryotic cells.

However, a very detailed study on the antiviral activity of lysozyme fragments obtained using proteolytic digestion with clostripain, which cleaves at the level of arginine residues in the C-terminal region, contains very interesting data [47]. This study highlights a nonapeptide, with the sequence RAWVAWRNR, which corresponds to residues 107–115 of the human lysozyme and appears to be the smallest lysozyme fragment capable of exerting antiviral activity (the prevention of infection and the inhibition of HIV viral replication) comparable to that of the intact lysozyme. Interestingly, this nonapeptide, which is present in the native lysozyme as an α-helix, belongs to a part of the lysozyme that is distinctly different from the catalytic site that exerts muramidase activity (Figure 4). It was later shown that this nonapeptide interacts in nanomolar concentrations with an HIV envelope glycoprotein that is critical for viral entry into cells. The study highlighted the nonapeptide component (two Trp residues separated by two others) that allows it to interact with the hydrophobic pocket of the gp41 glycoprotein and inhibit its fusion with the membrane of the cell to be infected [48]. An investigation of the mechanism of the antiviral effect showed that the nonapeptide modulates gene expression in HIV-infected cells, particularly of genes involved in survival and stress signaling. Similar effects were also demonstrated by transcriptomic analysis in eukaryotic cells exposed to intact HEWL [49]. This study documented the robust evidence on the modulation of genes associated with anti-inflammatory effects with relatively low doses (micromolar range) and short times of cell exposure (1 h).

Apart from the discussion on the part of the lysozyme molecule responsible for the antiviral activity, it is interesting to mention the activity of the recombinant CCR5 protein into which lysozyme has been incorporated. The transformed chemokine is able to exert a biphasic effect on HIV infection in a number of eukaryotic cell models: it weakly promotes infection at low doses (10^−10^–10^−12^ M) and markedly inhibits infection at high doses (10^−6^–10^−9^ M). The mechanism appears to be related to the downregulation of the wild-type CCR5 in eukaryotic cells [50].

## 4. Antiviral Activity: Disinfection over Norovirus Food Contamination

The first evidence for the effect of lysozyme on noroviruses comes from a paper published in 2015 by Takahashi et al., a research group that investigated this effect in detail [51]. Noroviruses usually infect humans through the consumption of food, often vegetables, although not exclusively, in the raw state or through direct contact with infected persons or their fluids, leading to gastroenteritis [52]. The work of Takahashi et al. emphasizes two things: the ability of lysozyme to eliminate contamination of food by the virus, and the fact that this activity comes from a component of lysozyme obtained after heat denaturation of the peptide. Using the murine norovirus model MNV-1, the mechanism of the antiviral effect of lysozyme subjected to heat denaturation was demonstrated. A classical heat denaturation of lysozyme was performed by dissolving lysozyme from hen egg white in 5% distilled water. The solution was filtered through a 0.20 μm filter, and the filtrate was heated to 100 °C in a bath for 40 min then cooled on ice for final preparation of the heat-denatured lysozyme.

Heat denaturation of lysozyme produces a chemical species that has completely lost its enzymatic activity but has increased antibacterial activity, including many Gram-negative bacteria. This activity is explained by the dimerization of the molecule and the exposure of an active site rich in tryptophan groups that allow interaction with the bacterial membrane, making the phospholipid membrane permeable and causing bacterial death [53]. It has also been suggested that the heat-denatured lysozyme is active against herpes simplex viruses in vitro, as its basic nature favors interaction with the viral particles [45,54]. The exposure of hydrophobic amino acids (residues 5–39, according to [51]) could also explain the antiviral activity. Indeed, these residues can interact with the structural proteins of the MNV1 virus, which contain many hydrophobic amino acids, and this interaction leads to inactivation of the virus [55]. However, there is no clear study in the literature to describe in detail the mechanism of action of the heat-denatured lysozyme in destroying capsid proteins and contributing to virus death.

The same research group had already demonstrated in detail, also using the murine norovirus MNS-1 model, the ability of the heat-denatured lysozyme to counteract contamination of fresh vegetables such as salads and their dressings, whereby the infectivity measured as PFU (Plaque Forming Units) was reduced by 2.6 or 4 logs compared to the untreated control [56]. Always using the MNV-1 mouse norovirus model, the containment of food infection by lysozyme was also demonstrated in some bread fillings, where treatment with the heat-denatured lysozyme immediately after MNV-1 inoculation completely eliminated viral contamination [57].

Heat-inactivated lysozyme has also been found to inactivate single-stranded RNA viruses that can cause hepatitis A (HAV). It is known that various foods, from seafood to strawberries, can be contaminated with HAV and cause hepatitis which, although not comparable to hepatitis B and C, can be very disabling [58,59]. Also in this case, a laboratory simulation with three HAV types with three different genotypes treated with a 1% solution of the heat-denatured lysozyme for 60 min showed a 2.6 log reduction in viral contamination compared to the control [60]. (For a general overview on use of lysozyme as a food preservative see Figure 5).

## 5. Antiviral Activity: Control of Foot-and-Mouth and Bovine Viral Diarrhea Virus Infections

The heat-denatured lysozyme has also been shown to counteract viral infections that cause foot-and-mouth disease (FMDV: Foot and Mouth Disease Virus) [61], a highly contagious viral infection in domestic and wild animals with cloven hooves, including cattle, water buffalo, sheep, goats, and pigs [62]. The study of Fukai et al. [61] showed how the heat-denatured lysozyme reduced the viral RNA load of FMDV O/Taiwan/1997 by up to 2.7 logs, and those of various other strains of the same virus were also reduced to varying degrees. Regarding the mechanism of action, a real-time RT-PCR test showed that the heat-denatured lysozyme destroys the capsid proteins of the virus and thus contributes to its death. As lysozyme is a protein of natural origin (the protein used in these experiments was obtained from hen egg white), another advantage of its use compared to conventional disinfectants is that it does not need to be removed during sterilization before food consumption and is completely safe for humans.

Lysozyme (hen egg white lysozyme) at concentrations ranging from 0.312 to 40 mg/mL was also tested on bovine diarrhea viruses, showing antiviral activity that depended on its presence throughout the course of virus infection (e.g., during the adsorption phase and the subsequent post-adsorption phase) and whose activity was directly proportional to the incubation period [63]. Of interest, in this study, is the strong antiviral activity of lysozyme in combination with lactoferrin, which suggests the use of this combination to combat this problematic disease, which is responsible for significant economic losses on livestock farms [64].

## 6. Antiviral Activity: Effects on Respiratory and Influenza Viruses

Type A influenza viruses, such as A/Texas/36/91 (H1N1), A/Qld/6/72 (H3N2), A/Ann Arbour (AA)/6/60 (H2N2), A/Beijing/352/92 (H3N2), have been shown to reduce the release of lysozyme from neutrophils in the airways (approx. 50% reduction compared to untreated controls) [65]. Apart from the effect on the lack of antibacterial activity (highlighted by the authors), we can also hypothesize that influenza viruses reduce the presence of lysozyme in the mucous membranes of the respiratory tract, as it may affect their infectious activity, as shown in a study analysing the antiviral effects of components of submandibular sublingual component, among which a certain amount of lysozyme is present [66]. The antiviral activity of saliva against influenza A and HIV viruses has been emphasized [67], also taking into account the different properties and potential of this fluid depending on the type of secretory gland [67]. The following is an overview of studies on the activity of lysozyme against different types of viruses involved in influenza and related respiratory diseases.

The research by Huang et al. [68] on influenza viruses, SARS-CoV and SARS-CoV-2 is based on the observation of the anti-viral activity of lysozyme denaturated with heat on noroviruses and HAV. They optimized the antiviral activity against H1N1 virus by subjecting the lysozyme to different heat denaturation conditions, varying the pH from 3 to 8, the heating time from 10 to 180 min and the concentration of denatured lysozyme tested on viruses. Using the optimized operating conditions, the same researchers then demonstrated antiviral activity, measured as inhibition of virus entry into the target cells, of the heat-denatured lysozyme against the avian viruses H5N1, H5N6 and H7N1 and also against the virus responsible for the COVID-19 pandemic, with the IC_50_ in the concentration range of ng/mL. The usefulness of lysozyme in upper respiratory tract infections was reported in a review analyzing the antimicrobial peptides present in respiratory secretions and reporting how lysozyme can combine antimicrobial activities with antiviral activities [69]. In a study in which lysozyme was formulated together with niclosamide for administration into the upper and lower respiratory tract via dry powder inhaler, nebulizer, and nasal spray, strong antiviral activity against MERS-CoV and SARS-CoV-2 was shown in vitro and in vivo, further highlighting that this formulation protects against secondary infections caused by methicillin-resistant *Staphylococcus aureus* pneumonia and inflammatory lung injury [70].

## 7. Antiviral Activity: Evidence of Activity in SARS-CoV-2 and Control of COVID-19 

The antiviral activity of lysozyme described in the previous sections, involving an interaction with anionic phospholipids or a direct binding to glycoproteins and glycolipids, has many similarities to that described for antimicrobial peptides produced and secreted by different cell types and proposed for the control of SARS-CoV-2 infection [71], with references cited herein. These peptides are also cationic (i.e., with a net positive charge), and many of them, similar to lysozyme, also possess immunostimulatory properties. Their antiviral activities, similar to those previously described for lysozyme, are in line with expectations for an antiviral drug that acts directly on viral functions (replication, gene expression, and/or protein processing) or blocks viral attachment and fusion by interacting with viral proteins or their receptors on host cells [72,73]. Antiviral peptides and antimicrobial peptides were also the subject of a detailed report on a series of clinical trials to combat SARS-CoV-2 infection and to attenuate COVID-19 (for exact details, the reader is invited to read [74]). Accordingly, C-lysozyme has been included in the group of antimicrobial peptides with antiviral activity that can potentially be used against SARS-CoV-2, as it deactivates the viral envelope and thus blocks entry into target cells [75]. Lysozyme is one of the components of whey proteins in human milk (approximately 0.4 g/L), and it has been suggested that milk has antiviral activity against SARS-CoV-2 [76,77]. This amount of lysozyme may also play a role in controlling SARS-CoV-2 infection, as it has been shown to decrease the production of various inflammatory cytokines such as IL-6, IL-8, IL-1β, TNF-α, and MCP-1 (the monocyte chemoattractant protein-1 induced by the spike proteins of SARS-CoV-2), as shown by Song et al. [78]. Nevertheless, the antiviral activity of the lysozyme contained in milk against SARS-CoV-2 continues to be emphasized, even if its individual role in the mixture of components present is not clear [79]. A detailed analysis of the properties of lysozyme that make it a candidate against SARS-CoV-2, or to limit the manifestations associated with SARS-CoV-2 infection, is reported in a comprehensive review by Mann and Ndung’u, which refers to its antiviral properties to prevent COVID-19-related infections, to its direct and/or immune-mediated antiviral properties, and also to its antioxidant properties [80], and references are cited herein.

However, the only study documenting the activity of lysozyme against SARS-CoV-2, using lysozyme from hen egg white, shows a significant activity when lysozyme is brought into direct contact with the virus at concentrations of 0.19 to 1.00 mg/mL, which are not toxic to Vero cells, and when the lysozyme–virus mixture is applied to the Vero cells at a second moment (the authors refer to this as pre-treatment and post-treatment, respectively). A clear antiviral activity is observed at the highest concentrations of 0.75 or 1.00 mg/mL, and it is clearly recognizable with the combination of pre-treatment + post-treatment and somewhat lower with the pre-treatment or post-treatment alone [81]. However, for the sake of clarity, it must be mentioned that the studies are at a preliminary stage, and there is still a great chance of receiving further insights.

## 8. Conclusions

According to the literature, lysozyme has a direct antiviral effect, although the exact mechanism of action still needs to be investigated. We focus on C-lysozyme and in particular on lysozyme from hen egg white, i.e., the most readily available lysozyme on the market, for which the most comprehensive and complete analysis of its chemical, biological, and even pharmacological properties has been certainly conducted. The available studies on antiviral activity are relatively limited, but fortunately, they are often consistent both in terms of hypotheses and results obtained in the in vitro models of viral infections with animal and human infectious viruses (Table 1). To summarize, lysozyme at relatively low concentrations is able to protect food from contamination by noroviruses, and its action is preferable to many substances used for this purpose, since lysozyme does not need to be removed from the treated food before consumption, is effective against viruses infecting cloven-hoofed domestic and wildlife animals, and is also effective against influenza and respiratory viruses in humans. 

One hypothesis on the mechanism of the antiviral effect is that it depends on the characteristics of the chemical structure of the lysozyme and in particular on its positive charge, independent of its enzymatic activity on the peptidoglycan. In this context, the study on the heat inactivation of lysozyme is important, showing the loss of antibacterial activity against Gram-positive bacteria and the acquisition of antibacterial properties against Gram-negative bacteria and the direct antiviral activity on viral cells, resulting in changes that prevent the attack and penetration of viruses into the host cell. The significance of these studies lies primarily in the fact that they show that lysozyme has a direct effect against viruses. However, it is clear that further studies are needed on the exact mechanism of this action, and the demonstration of having identified an amino acid sequence (the nonapeptide, RAWVAWRNR) that reproduces the antiviral effect of the whole molecule represents an important step forward.

In summary, the analysis of the literature data on the antiviral activity of lysozyme in this review shows that lysozyme has a direct effect against various types of viruses that threaten human health. The direct antiviral activity is important as it complements the immunostimulatory properties of lysozyme [17], suggesting that this natural peptide may play pharmacological and therapeutic roles in various pathological conditions where viral infections are the main cause. In this context, the fact that SARS-CoV-2 is also sensitive to the activity of lysozyme is interesting, albeit preliminary. Considering the ease of the oral administration of lysozyme, also in combination with other therapies, if these data are confirmed by a more detailed analysis, it could be hypothesized that lysozyme could also be used to mitigate the manifestation of COVID-19.

## Figures and Tables

**Figure 2 molecules-29-00652-f002:**
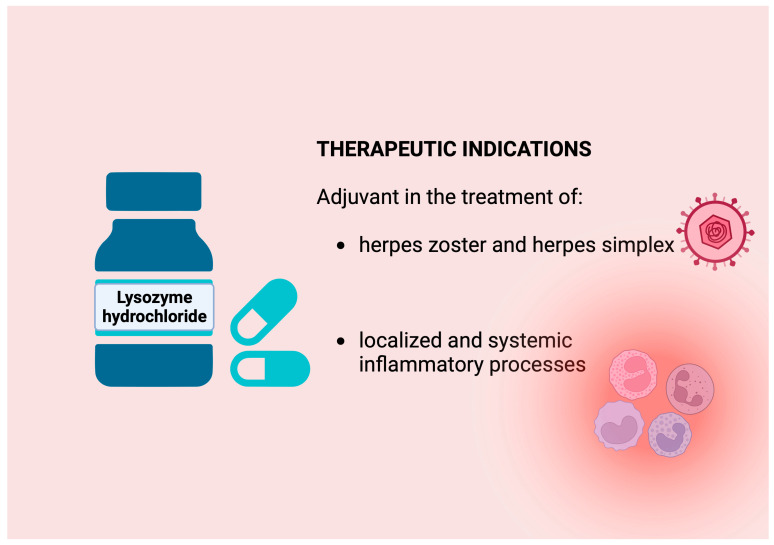
Therapeutic uses of hen egg white lysozyme. HEWL is the only lysozyme also available in a pharmaceutical form and used for therapeutic purposes as an antiviral for systemic use.

**Figure 3 molecules-29-00652-f003:**
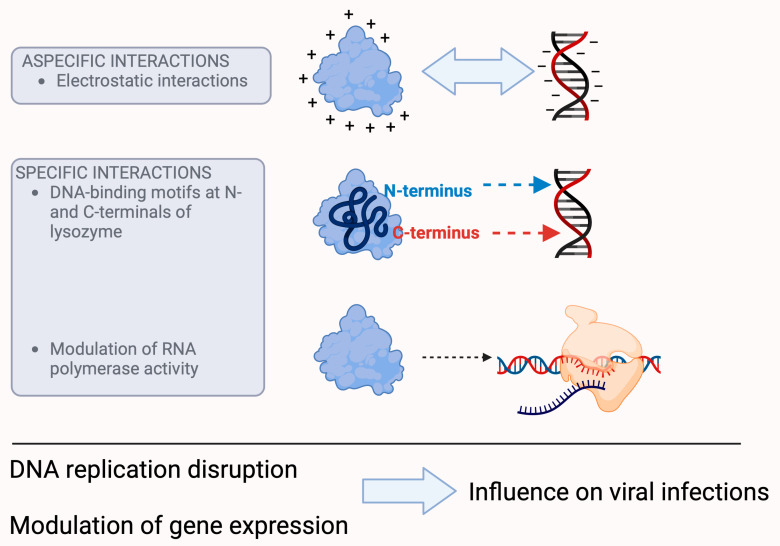
Presumed mechanisms of the antiviral effect of lysozyme. Lysozyme is able to interact with nucleic acids because its positive charge alters the charge and electrophoretic mobility of DNA, which affects transcription, translation, and cell division. The latter may also be the result of specific interactions, as lysozyme has DNA-binding motifs at the N- and C-terminals of its molecule. In addition, lysozyme is able to control the activity of RNA polymerase. All these processes can ultimately lead to the control of viral infectivity. (For further information about the hypotheses of the antiviral effect of lysozyme, please refer to [29,33,34].)

**Figure 4 molecules-29-00652-f004:**
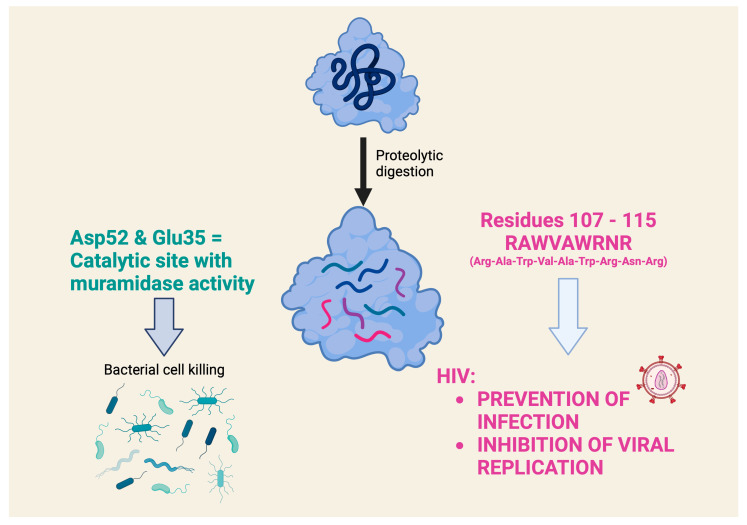
What role does the chemical structure of lysozyme play in antiviral activity? The antiviral activity of several lysozymes is attributed to a mechanism of action independent of muramidase activity, which produces the lytic antibacterial activity. A nonapeptide (RAWVAWRNR), which corresponds to residues 107–115 of the human lysozyme, appears to be the smallest lysozyme fragment that can exert an antiviral effect. This nonapeptide belongs to a part of the lysozyme that is clearly distinct from the catalytic site that exerts muramidase activity [45,47].

**Figure 5 molecules-29-00652-f005:**
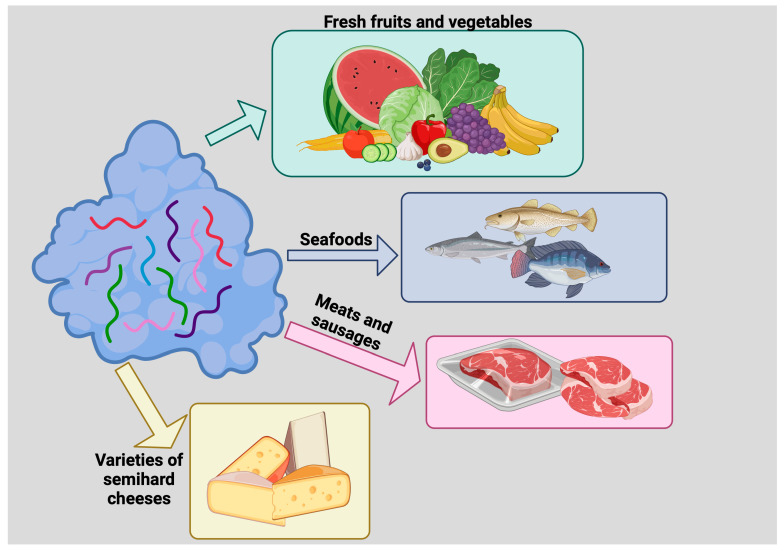
Use of lysozyme as a food preservative. Lysozyme inhibits the growth of deleterious microorganisms, thus prolonging food shelf life. Lysozyme has been used to preserve fresh fruits and vegetables, seafoods, meats and sausages, and varieties of semihard cheeses such as Edam, Gouda, and some Italian cheeses.

**Table 1 molecules-29-00652-t001:** Antiviral activity of lysozyme.

Lysozyme and Model	Antiviral Activity	Reference
C-lysozyme purified from different sources against HIV cell infection	Anti-HIV activity assessed using HIV-1 core protein p24 expression in chronically infected ACH-2 lymphocytes and U1 monocytes	[31]
HEWL tested on HIV infectivity	EC_50_ 55nM; preferred activity before HIV infection; restricted HIV attachment to host cell CD4	[82]
C-lysozyme overexpression in Bombix mori	Inhibition of cell proliferation of B. mori nucleopolyhedrovirus	[24]
Lysozyme heat-treated for 40 min at 100 °C tested against MNV-1, a surrogate of human norovirus	4.5 log reduction in infectivity of norovirus	[51]
1% heat-denatured HEWL added to different types of salads exposed to murine norovirus-1 (MNV-1)	General decrease in viral infectivity and inactivating effect on MNV-1	[56]
1% heat-denatured HEWL for 60 min against three strains of hepatitis A viruses and murine and human noroviruses	Potent virus inactivating activity and disinfectant action for fruits	[57]
Addition of 1% heat denatured lysozyme against murine norovirus-1 inoculated into chocolate cream or marmalade jam	Decrease in infectivity by 1.2 log PFU/g in chocolate cream and by 0.9 log PFU/g in marmalade jam	[60]
Heat-denatured lysozyme mixed and plated with foot-and-mouth disease virus at room temperature for 1 min	Inhibition of virus infectivity and of RNA loads	[61]
CCR5-T4-Lysozyme on HIV-1 infection in THP-1 cell lines, human macrophages, and PBMCs from clinical isolates	Inhibitory activity at the higher concentrations	[50]
C-lysozyme contained in saliva of patients tested against H1N1 influenza virus (IAV) on Madin- Darby canine kidney cells	C-lysozyme contributes to the high anti-IAV activity of sub-mandibular sub-lingual saliva	[66]
Different heat inactivating conditions of HEWL against H1N1, H5N1, H5N6, and H7N1 influenza viruses and SARS-CoV and SARS-CoV-2	Inhibition of viral entry into target cells in the ng/mL range	[68]
Lysozyme combined with lactoferrin against bovine viral diarrhoea virus	Strong antiviral effect at dosages lower than those of the single drug	[63]
Heat-treated lysozyme against SARS-CoV-2 in vitro	Highest activity when lysozyme was pre-incubated with the virus and re-added to already-infected cells	[81]

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
