# Peer review of "Lysozyme: A Natural Product with Multiple and Useful Antiviral Properties"

_molecules, 2024, doi:10.3390/molecules29030652_

Round 1

Reviewer 1 Report

Comments and Suggestions for Authors

Thank you for your work. I would like to add a few comments that will help improve the quality of the manuscript.

1. Can you provide more specific examples to support your arguments, in particular with regard to the anti-virus activities and mechanisms in discussion? (3. Chemical structure and anti-viral activity).
 2. To increase the relevance of the publication, the importance of each study or finding should be emphasised.
3. An additional figure should be included to show the relationship between chemical structure and anti-viral activity. Figure 4 should be revised and corrected.

4. Section 5: The importance of lysozyme in the prevention and control of viral infections, such as foot-and-mouth disease and bovine viral diarrhoea virus, should be made known.   For the clarity of the presentation, please refer to the viruses in focus.
5. To improve the quality of the article, please briefly review the study by Fukai et al. highlighting the main findings on the reduction of viral RNA load.
6.  The mechanism of action of heat-denatured lysozyme in destroying capsid proteins and contributing to virus death should be clarified.  Relate the information on the mechanism of action to the benefits of naturally occurring lysozyme.

7. Conclude by hypothesising that, if the preliminary results are confirmed by more detailed research, lysozyme could be considered as a viable option to mitigate the effects of COVID-19, and could be used for the development of pharmaceutical formulations. ETC.

Comments on the Quality of English Language

Minor editing of English language required

Author Response

  1. The review already contains several concrete examples that support the description of the antiviral activity of lysozyme. The main objective was not to analyze a structure-activity relationship (QSAR), but rather to present the data and information supporting the antiviral activity of lysozyme from a biological and pharmacological point of view.
  2. The review presents, analyzes and discusses each published work taken into consideration without any priority on one of them. We do not understand what the reviewer wants with “the importance of each study or finding should be emphasised”.
  3. See response to question 1
  4. The review already refers to the importance of lysozyme in the prevention of viral infections such as foot-and-mouth disease and bovine viral diarrhea, which are discussed in the text and also highlighted in Table 1.
  5. The study by Fukai is cited with the main findings highlighted “The study of Fukai et al., [61] showed how the heat-denatured lysozyme reduced the viral RNA load of FMDV O/Taiwan/1997 by up to 2.7 logs, and those of various other strains of the same virus were also reduced to varying degrees. Regarding the mechanism of action, a real-time RT-PCR test showed that heat-denatured lysozyme destroys the capsid proteins of the virus and thus contributes to its death.” We do not believe to add any further comment. However, we have added the effect on the RNA load also in Table 1.
  6. The suggestion is quite interesting. However, there is no clear study in the literature to describe in detail what the reviewer wants to add here.
  7. That lysozyme could be an option to mitigate the effects of CoViD-19 is certainly an intriguing hypothesis. However, the studies supporting this hypothesis are currently still at a very very preliminary stage and this conclusion seems immature, to say the least. The results of future studies may provide new and expanded insights into this potential use of lysozyme. 

Reviewer 2 Report

Comments and Suggestions for Authors

Should be revised.

Comments on the Quality of English Language

Minor mistakes, typo-errors needs to rectified.

Author Response

  1. A sentence reporting the novelty of the present review over the past works is now included.
  2. The illustrations are intentionally general and generic; they are meant to summarize in a simple and immediate way what the text tells in more detail, in order to arouse the reader's interest and curiosity so that he can consult the bibliography cited.
  3. The review already contains several concrete examples that support the description of the antiviral activity of lysozyme. The main objective was not to analyze a structure-activity relationship (QSAR), but rather to present the data and information supporting the antiviral activity of lysozyme from a biological and pharmacological point of view.
  4. Not clear what the reviewer requested
  5. The review carefully reports and discusses any clearly and scientifically sustained report on the possible structure of lysozyme responsible for the pharmacological effect. Data on this aspect are often very poor of details and we decided to use only those for which repeated confirmations were described.
  6. A careful bibliographical search was carried out and the sources cited are those relevant to a comprehensive description of the antiviral activity of lysozyme. Other articles on lysozyme, although recent, were not considered useful for this purpose as they focused on aspects other than biological or pharmacological activity.

Minor comments

Accepted all